# Optimizing the Quality and Commercial Value of Gyokuro-Styled Green Tea Grown in Australia

James Krahe [1,2] and Michelle A. Krahe [3,*]

1    School of Environmental and Life Sciences, University of Newcastle, Ourimbah, NSW 2258, Australia; james.krahe@fial.com.au
2    Food and Agribusiness Growth Centre, Brisbane, QLD 4108, Australia
3    Health Group, Griffith University, Gold Coast, QLD 4222, Australia
*    Correspondence: m.krahe@griffith.edu.au

**Abstract:** Gyokuro is a style of Japanese green tea produced by employing agricultural shading in the weeks before harvest. This method results in a tea product with different organoleptic and chemical properties than common Japanese green tea. In an effort to yield the highest quality and commercially valuable green tea product, the present study explores the influence of shading treatments and the duration of shading on the natural biochemistry of the green tea plant. This study applied shading treatments at light intensity conditions of 40%, 16%, 10% and 1% of available ambient light and the application of a red-colored shade cloth of 60% opacity. The Quality Index Tool was used to measure the quality and commercial value of the green tea, using individual target constituents (theanine, caffeine and the catechins) quantified from HPLC analysis. This study shows that very high levels of total visible spectrum light shading (~99%) is required to achieve improvements in quality and commercial value. Specifically, this improvement is a direct result of changes in the mood- modifying bioactive metabolites theanine and caffeine. This study concludes that in green tea growing regions with more hours of sunlight per year, such as on the Central Coast of Australia, more intense shading will achieve products with improved quality and commercial value, which has more potential to be marketed as a functional ingredient.

**Keywords:** green tea; shading; quality; commercial value; functional food

## 1. Introduction

Light is one of the most important factors that influences plant growth. Gyokuro is a high-grade style of green tea typically prepared by shading the plant for about three weeks before harvesting, causing the leaves to endure less photosynthesis, keeping strong flavored amino acids in the plant, and giving Gyokuro its fuller taste [1]. This process results in increased concentrations of theanine, the most abundant free amino acid in green tea, and caffeine in the leaves and a tea with a strong umami taste [2,3]. Consequently, Gyokuro-styled teas are much more expensive compared to the more commonly consumed Sencha-styled green teas, which are grown under full sun. The first and most noticeable visual difference is the color. Sencha's are typically lighter green to yellow, while Gyokuro is typically a deep green due to increased chlorophyll concentrations within the leaf [1,4,5], caused by the plant's response to the reduced amount of sunlight available—the plant attempts to increase chloroplast numbers so that it can improve its photosynthetic efficiency to output levels similar to those before the shading treatment [6,7]. It is further speculated that this physiochemical response to natural light is not caused by a reduction in the complete spectrum of light, but only by reduced access to wavelengths that the plant would be actively absorbing [8,9]. Therefore, as green wavelengths are reflected, wavelengths associated with the colors red and blue are absorbed [10]. It is hypothesized that the Camellia *sinensis* var. *sinensis* plant increases the concentration of chlorophyll as a direct

effect of the loss of red wavelengths along the spectrum of natural light. In a recent study conducted in Mississippi, United States, also an emerging tea growing region, red shading of green tea plants was considered most advantageous for improving plant growth and leaf quality by increasing the content of theanine and free amino acids when compared to no-shade control [11].

Similar to other shaded agricultural products, Gyokuro is expensive and produced in lower volumes due to the increased costs involved in growing the plants. In Japan, the tradition is to use up to 90% shading during the production of Gyokuro [1,12]. However, 60% shading has been reported to be the most beneficial [13]. In addition, the optimal period of green tea shading ranges from 4 to 28 days, differing by locality, shade intensity and time within the growing season. Where green tea is grown in Australia, the conditions have been selected to match those in established tea growing regions of Asia, particularly those of the Shizuoka region of Japan. This is a major tea-producing region, renowned for producing very high-quality green teas, with well-established and documented growing conditions [14]. Despite the green tea market in Australia being relatively small compared to other Asian countries, the Australian green tea industry may look towards the export market of Japan or China as a key avenue to high-value sales. For example, the end of year and New Year celebrations in Japan coincide with the peak sales periods for Australian green tea in Japan (December to January) and there are Japanese winter months during which fresh locally produced green tea is not available. Conditions that affect high-quality green tea production include the levels of solar radiation, daily temperatures and humidity, water availability and soil drainage [15]. Further, to optimize the quality of Gyokuro-styled teas, the bioactive constituents of theanine, caffeine and the catechins within the leaves of the Camellia *sinensis* var. *sinensis* plant are important. The concentration of these constituents can vary depending on cultivation conditions such as the harvest season, sunshine conditions, breeds of the tea plant and geography [16]. Changes in the concentration of these constituents can markedly change the taste sensory profile and enjoyment of the tea; if the flavor of the beverage is poor, then the quality and sales of the product will be affected. For Japanese Gyokuro-styled teas, a delicate balance between the sweet-salty sensation of umami, perceived from the theanine, and the bitterness and astringency of the caffeine and catechins is desirable [2,17]. Ratios of these constituents have been associated with the retail value of the final tea product [18]. Objective methods for the determination of green tea quality must be developed so that subtle variations caused by specific agricultural and processing methods can be identified.

The primary objective of this study was to determine the optimal growing conditions of Gyokuro-styled tea grown on the New South Wales Central Coast of Australia. Specifically, this includes the optimal shade intensity, solar intensity and duration of shade treatment to produce the highest quality and commercial value tea. Secondly, it is anticipated that the findings from this study will provide useful information about the functional properties associated with optimizing the production of Gyokuro-styled green tea in Australia.

## 2. Materials and Methods

### 2.1. Cultivation of Plants

All green tea (Camellia *sinensis* var. *sinensis*) plants used in this study were cultivated from a site on the New South Wales Central Coast of Australia (see detailed description in [19]. The climatic conditions of the Central Coast of Australia are similar to those of the Shizuoka region in Japan—except for a difference in the number of sunlight hours. Gosford receives approximately 18% more hours of sunlight per year than the Shizuoka region [14]. Established Yabukita varietals were used in this study and control and treatment plants within the same row were used. This ensured they were exposed to similar environmental factors (i.e., sunlight intensity and direction) as well as agricultural conditions, including irrigation. The only variable was the experimental factor of natural light shading intensity.

## 2.2. Experimental Design

Treatment plants received a direct application of agricultural grade shade cloth of specific opacity, number of layers and color. The shade cloth was applied to a frame close but not touching the plant to control for any stress, constriction or potential damage placed upon the plant by the presence of the shade cloth. Figure 1 shows the application of shade cloth to green tea plants in single rows. The level of ambient light (i.e., luminous emittance (lx)) was measured using a Testo 540 handheld Pocket Lux Meter (Testo Pty Ltd., Croydon South, Australia) to achieve the targeted shading conditions.

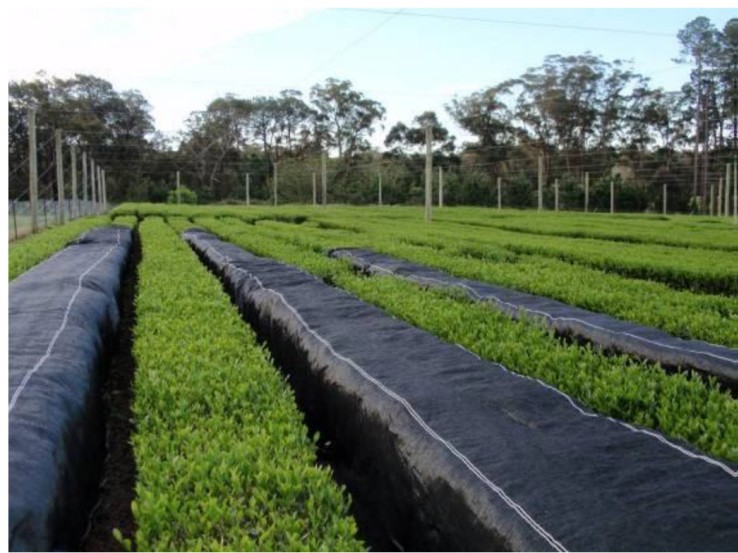

**Figure 1.** Application of shade cloth to three rows of Camellia *sinensis* var. *sinensis* plants, with a single layer of 90% black shade cloth applied to the row covering the top and the side surfaces.

Site randomization and blocking were utilized for all treatment and control plants, as plants within a row shared common constant pressure irrigation and fertilizer application line. The only difference in the shade cloth material were those specifically intended to test specific hypotheses.

### 2.2.1. Determination of Optimal Shading Intensity and Color

Five groups of six plants were randomly selected for each experimental condition, from a single row of 65 mature and healthy Yabukita green tea plants. The optimal shading intensity was determined by testing a 60% and 90% opacity of black shade cloth application in both single and double layers. This created light intensity conditions of 40%, 16%, 10% and 1% of available ambient light. Testing the specific color of the shade cloth involved the application of a red-colored shade cloth of 60% opacity to six randomly selected plants within a single row.

A control group received no shading (i.e., receiving 100% of the ambient light). The shading period was 14 days prior to a normal pre-planned harvest, after which samples were collected and analyzed. Analysis of data was based upon the percentage of available ambient light intensity.

### 2.2.2. Determination of Optimal Duration of Shading

Agricultural grade shade cloth at 90% opacity was applied to four plants within a single row receiving constant pressure irrigation enrichment. Shading was conducted over 11 weeks including the first two harvests of the harvesting year. During this period plants received a minimum of six hours of direct sunlight and an additional four hours of reflected sunlight per day. During this time, samples from each of the plants were collected and

analyzed weekly. Analysis of data was conducted based on shaded treatment group versus control, for leaf structural morphologies.

### 2.3. Collection and Preparation of Samples

Samples were collected using a hand-harvesting method. Five cuttings were collected from each of the control and treatment plants and taken from the new growth, which consisted of the first five leaves. All samples were packaged into high-moisture and light-barrier laminate packaging and transported to the University of Newcastle (Ourimbah, Australia) for analysis. The green tea samples were stored in cool, dry and dark ambient conditions before being prepared into green tea infusions. Green tea samples were prepared in triplicate for each treatment group.

### 2.4. Analysis of Samples Using HPLC and Standard Chromatographic Conditions

Samples were analyzed using a high-performance liquid chromatography (HPLC) system (Shimadzu Analytical Australia, Rydalmere, Australia), controlled via an SCL-10A VP control unit and the Class-VP 5.03 software. This method is detailed in Krahe et al. [19]. In short, the HPLC method was adapted from validated methods for the separation of the different green tea metabolites [20], and for the identification of theanine in green tea [21]. This method includes the identification and measurement of the major green tea constituents theanine, caffeine and the catechins (EC, (-)-epicatechin; ECG, (-)-epicatechin-3-gallate; EGC, (-)-epigallocatechin; EGCG, (-)-epigallocatechin-3-gallate; GCG, (-)-gallocatechin gallate) and controlled with the IS L-tryptophan. The quantitative analysis was completed in triplicate for each sample and the concentration of each constituent in the form of mg of constituent per g of dried tea was calculated.

### 2.5. Calculating Quality

The Quality Index (QI) Tool is a validated measure of the quality and commercial value of green tea (see [18] for full details). Individual target constituents being theanine, caffeine and the catechins (EC, ECG, ECG, EGCG and GCG) were quantified from HPLC analysis and used to assess the five quality markers of green tea (total catechins, theanine:total catechin, theanine:caffeine, ECGC:EGC, EGCG:GCG), of which the significance on the organoleptic properties of green tea are previously detailed [18]. The corresponding quality indicators (quality score and commercial value) were identified (Table 1).

**Table 1.** The Quality Index (QI) Tool. The quality equation for each quality marker is used to determine quality indicators for green tea.

| Quality Indicator | |
|---|---|
| **Quality Score** [1] | **Commercial Value** [2] |
| >3.5 | Very High |
| 3.0 | High (>400) |
| 2.5 | Medium—High |
| 2.0 | Medium (200–400) |
| 1.5 | Low—Medium |
| 1.0 | Low (>200) |
| <0.5 | Very Low |

[1] The average quality score is rounded to the nearest half unit; [2] Commercial value is defined as the market value of green tea where: high—AUD 500/kg; medium—AUD 300/kg; low—AUD 150/kg. Adapted from [16].

### 2.6. Statistical Analysis

All statistical analyses were conducted using SPSS 26.0 Statistics for Windows (IBM Corp 2021, Armonk, NY, USA). Mean $\pm$ SD are presented, and the significant level was established at $p \leq 0.05$. Multivariant analysis was used to determine the preservation effect of growing conditions for each target constituent, with Bonferroni post hoc analysis to determine individual differences between groups.

## 3. Results

In an effort to yield the highest quality and commercially valuable green tea product, the present study provides insight into the influence of shade treatment and the duration of shading on the natural biochemistry of the green tea plant. The effect of reducing the available ambient light intensity significantly ($p < 0.05$) improved the quality score of green tea products grown at 1% light intensity, compared to all other light intensities, including 100% (control), 40%, 16% and 10% (Table 2). This improvement was primarily due to a significant increase in the quality marker for theanine:caffeine (w:w).

**Table 2.** Quality markers and quality indicators of green tea plants grown under various ambient light intensities.

| Treatment | Quality Markers | | | | | Quality Indicators | |
|---|---|---|---|---|---|---|---|
| Light (%) [Mean lx] | TC (mg/g) [1] | T:TC (w:w) | T:C (w:w) | EGCG:EGC (w:w) | EGCG:GCG (w:w) | Quality Score | Commercial Value |
| 100 [2] [42, 319] | 94.4 ± 5.2 [a,b] | 0.6 ± 0.0 [a] | 1.2 ± 0.1 [a] | 3.0 ± 0.4 [a] | 8.3 ±1.8 [a,b] | 2.0 ± 0.1 [a] | Medium |
| 40 [16, 566] | 95.9 ± 2.6 [a,b] | 0.5 ± 0.0 [a] | 1.2 ± 0.1 [a] | 3.0 ± 0.3 [a] | 6.8 ± 0.7 [a,b] | 1.9 ± 0.1 [a] | Low—Medium |
| 16 [6, 695] | 98.5 ± 4.1 [a,b] | 0.6 ± 0.0 [a] | 1.3 ± 0.1 [a] | 3.0 ± 0.5 [a] | 6.4 ± 0.4 [a] | 2.0 ± 0.2 [a] | Medium |
| 10 [4, 186] | 99.6 ± 4.3 [a] | 0.5 ± 0.0 [a] | 1.2 ± 0.1 [a] | 3.1 ± 0.3 [a] | 9.1 ± 1.5 [b] | 1.9 ± 0.1 [a] | Low—Medium |
| 1 [402] | 89.6 ± 8.8 [b] | 0.7 ±0.1 [a] | 1.6 ±0.1 [b] | 3.5 ± 0.8 [a] | 6.1 ±1.9 [a] | 2.4 ± 0.1 [b] | Medium—High |

[1] Proportion of dry tea; [2] Indicates the control group. Mean ± SD values for the quantified values of theanine, caffeine and catechins (EG, EGC, EGCG, GCG and ECG) from the leaves of green tea plants. The lx = luminous emittance measured at time of setup; TC = Total Catechin; T = Theanine; C = Caffeine. Values with shared superscripts within the same column indicate a non-significant difference in value ($p > 0.05$). Table S1 lists the quantification of the target constituents theanine, caffeine and total catechins.

The effect of using red-colored shading to reflect the red wavelengths of light, compared to ambient sunlight intensity, did not significantly influence quality indicators. However, the loss of the red wavelengths of visible light caused a significant increase ($p < 0.05$) to the total catechin content, and significant reductions to the theanine:total catechin and EGCG:GCG ratios, compared to reduced total visible and ambient light conditions (Table 3). A possible cause for this increase might be attributed to the plants' natural reaction to specific wavelengths of light available. This treatment reduces the availability of red-light sources, leaving greater proportions of higher wavelength intensities associated with blue light, shown to encourage development and leaf bulk, as well as chloroplast migration in some plant species [22,23]. The black-colored shading did not affect the concentration of target constituents (see Table S2).

After 11 weeks of shading, total catechin concentration was significantly reduced, and from week three theanine:caffeine ratios significantly increased ($p < 0.05$), compared to the control (Figure 2). At week 11, the EGCG:GCG ratio also significantly improved with reduced light conditions (90% opacity) (see Table S3). The quality score for plants grown under 10% ambient light conditions slightly improved, compared to the control, and a slight decrease in quality was observed at week five in the shaded group, albeit this was not significant (Table 4).

**Table 3.** Quality markers and quality indicators of green tea plants grown with and without red-colored wavelengths compared to full ambient light conditions.

| Treatment | Quality Markers | | | | | Quality Indicators | |
|---|---|---|---|---|---|---|---|
| Shade Color [Mean lx] | TC (mg/g) [1] | T:TC (w:w) | T:C (w:w) | EGCG:EGC (w:w) | EGCG:GCG (w:w) | Quality Score | Commercial Value |
| NA [2] [41, 236] | 94.4 ± 5.2 [a] | 0.6 ± 0.0 [a] | 1.2 ± 0.1 [a] | 3.0 ± 0.4 [a] | 8.3 ±1.8 [a] | 2.0 ± 0.0 [a] | Medium |
| Black [15, 900] | 89.0 ± 4.5 [a] | 0.6 ± 0.0 [a] | 1.2 ± 0.1 [a] | 2.6 ± 0.3 [a] | 7.6 ± 1.6 [a] | 2.1 ± 0.2 [a] | Medium |
| Red [16, 125] | 113.5 ± 12.5 [b] | 0.4 ± 0.0 [b] | 1.3 ± 0.1 [a] | 2.4 ± 0.5 [a] | 4.2 ± 1.7 [b] | 1.9 ± 0.2 [a] | Low—Medium |

[1] Proportion of dry tea; [2] Indicates the control group. Mean ± SD values for the quantified values of theanine, caffeine and catechins (EG, EGC, EGCG, GCG and ECG) from the leaves of green tea plants. The lx = luminous emittance measured at time of setup; TC = Total Catechin; T = Theanine; C = Caffeine. Values with shared superscripts within the same column indicate a non-significant difference in value ($p > 0.05$). Table S2 lists the quantification of the target constituents theanine, caffeine and total catechins.

**Table 4.** Quality indicators of green tea plants grown in reduced light intensity (90% opacity) compared to full ambient sunlight conditions over 11 weeks.

| | Quality Indicator | | | |
|---|---|---|---|---|
| Week | Light Intensity (100%) [1] | | Light Intensity (10%) | |
| | Quality Score | Commercial Value | Quality Score | Commercial Value |
| 0 | 2.2 ± 0.0 | Medium | 2.2 ± 0.1 | Medium |
| 1 | 2.2 ± 0.0 | Medium | 2.2 ± 0.1 | Medium |
| 2 | 2.2 ± 0.1 | Medium | 2.2 ± 0.1 | Medium |
| 3 | 2.1 ± 0.1 | Medium | 2.1 ± 0.1 | Medium |
| 4 | 2.1 ± 0.1 | Medium | 2.1 ± 0.1 | Medium |
| 5 | 2.2 ± 0.1 | Medium | 2.2 ± 0.1 | Medium |
| 6 | 2.1 ± 0.0 | Medium | 2.1 ± 0.1 | Medium |
| 7 | 2.1 ± 0.1 | Medium | 2.1 ± 0.1 | Medium |
| 8 | 2.0 ± 0.1 [a] | Medium | 2.1 ± 0.1 | Medium |
| 9 | 1.9 ± 0.1 | Low—Medium | 2.1 ± 0.1 | Medium |
| 10 | 2.0 ± 0.1 | Medium | 2.0 ± 0.1 [a] | Medium |
| 11 | 1.8 ± 0.0 | Low—Medium | 2.0 ± 0.1 [b] | Medium |

[1] Indicates the control group. Mean ± SD values. [a] First instance when a quality indicator is significantly different ($p < 0.05$) compared to baseline (week 0); [b] Value is significantly different ($p < 0.05$) to control group.

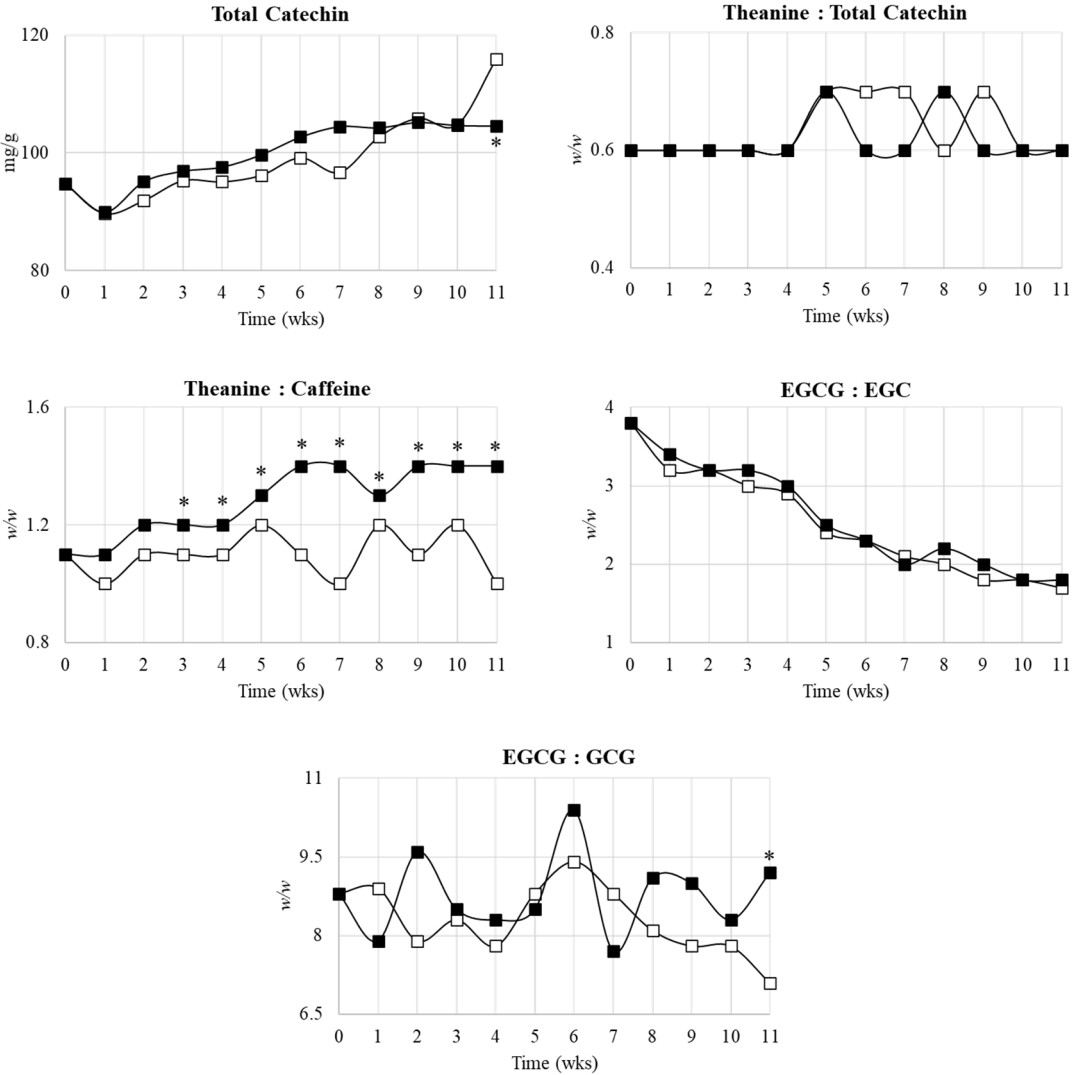

**Figure 2.** Line graph of mean effect on the concentration of quality markers of green tea grown in different light conditions over 11 weeks. An asterisk indicates a significant difference ($p < 0.05$) between plants grown in reduced light intensity (black square: 90% opacity) vs. full ambient sunlight (white square) at the same study week.

## 4. Discussion

This study examines the effect of agricultural shading of green tea grown in Australian conditions to produce optimal quality Japanese-styled Gyokuro green tea products. We found that the highest quality and commercially valuable tea was produced by reducing ambient light intensity to one per cent, for up to two weeks. Adjusting the shade color or reducing light intensity over an extended period did not significantly influence the quality indicators of green tea but did alter some quality marker profiles. This suggests that due to the greater number of sunlight hours in Australia, compared to the traditional growing locations in Japan, very intense shading needs to be utilized to achieve the quality improvements.

The quality improvement observed in this study was owing to an increase in the concentration of theanine (and subsequently the theanine: caffeine), as a result of shading to one per cent ambient light intensity. This produced a tea that is more desirable to consumers, with greater levels of theanine, an improved umami flavor and lower levels of caffeine, which is associated with bitterness [24]. This change may be partially explained by the plant's reduced potential for active photosynthesis. During periods of limited available

sunlight, the plant will slow its metabolism and concentrate theanine, which is one of its free amino acid resources [24]. This is because the theanine movement process is energy passive in association with transpiration, and, as such, the movement of theanine continues at a similar rate with or without the plant actively photosynthesizing [25,26]. The influence of light intensity treatment on theanine concentration and subsequent improvement in the quality and commercial value of green tea aligns with other reports. Hirai et al. [27] show that improvements in green tea quality may be achieved using 60–99% opacity shade cloths. In comparison, the work of Saijo [13] indicates improvement at 10% and 40% ambient light intensity. While these results imply that a less intensive level of shading is needed (compared to that observed in our study), consideration must also be given to the geographic and climatic differences (especially the ambient solar energy) between tea-growing regions. For example, during the peak period of the Australian summer, the energy from solar radiation reaching the Central Coast region is ~23 MJ/m$^2$/day, compared to ~11 MJ/m$^2$/day in the region of Shizuoka, Japan [14]. The length and intensity of daily solar radiation, daily temperature and humidity, water availability and soil drainage are also important to ensure that the plant does not enter long periods of growth dormancy [15]. Investigation of other influencing factors is necessary, and although high opacities for optimal shading conditions to produce Gyokuro products are predominantly favorable, the sunlight intensity is not uniform, or may be influenced by other factors such as air pollution levels [28,29]. A limitation of this study is that the light intensity for each of the treatments was only determined at the commencement of the study and not continuously measured throughout. The initial study design was to determine the outcome of the agricultural shading practice upon green tea quality, but further work detailing the mechanisms of change caused by the reduction of light and specific wavelength would also be useful for the understanding of causality.

Another consequence of altered shading conditions was an increase in the concentration of total catechins, an important and prominent group of compounds that affect the flavor, appearance, aroma and quality of the tea [17]. According to Xia and Gao [30], the catechins, in particular EGCG, can provide important antioxidant protection from free radical damage to chlorophyll and chloroplasts. Hence, the presence of catechins within the Camellia plant has been suggested to improve the efficiency and functioning of photosynthesis, thereby ensuring rapid growth and development during the summer months [23]. It has also been speculated that the antioxidant catechins are highly stable within the tea plants' natural matrix, allowing for other compounds to concentrate within the leaves [31].

Theanines and catechins are important in tea quality but also provide functional food properties [24,32]. Currently within the food industry, functional foods can produce additional revenue for manufacturers, as consumers try to gain more nutrition and benefits from their foods. This desire has resulted in new consumer products and marketing aimed at food with health benefits in addition to basic nutrition [33,34]. The most widely studied and regarded health benefit of theanine is the role it plays in mood and concentration [35]. It is well accepted that theanine can induce relaxation without drowsiness, helping to focus the mind, due to the stimulation of Gamma Amino Butyric Acid (GABA), serotonin and tryptophan production [36,37]. The combination of these hormones results in relaxation and focus and an increase in alpha brain waves, which improves cognition. Theanine can also improve blood pressure via relaxation and thereby reduce stress-induced hypertension [37]. However, the effect of theanine is in some ways balanced by the presence of caffeine in green tea, which despite its positive effect on concentration, is a nervous system stimulant and thus can counter the calming effect of theanine [38]. Shading, due to its capacity to modulate theanine and caffeine, could therefore be lucrative to those promoting theanine as a functional food found naturally in green tea. Similarly, examples of the nutraceutical and functional food health claims related to green tea catechins include their anticancer benefit, effect on lipid profile and management of hyperlipidemia, protection against UV-induced damage due to erythema inflammation and weight loss [39–42]. The beneficial

effects ascribed to tea catechins are thought to be derived from their ability to scavenge reactive oxygen species and function indirectly as antioxidants through their effects on transcription factors and enzyme activities [43,44]. To fully understand the contribution green tea constituents make to human health, larger studies related to consumption of the beverage in Western diets (which can be limited and sporadic) are warranted.

Despite the establishment and cost of shading being much greater for longer-term shading compared to short-term, very low light intensity shading, the benefits seen in this trial should be encouraging for producers. This is because the resulting shaded teas are more commercially valuable products and provide greater commercialization opportunities for producers. These teas contain higher concentrations of mood-modifying bioactive metabolites, theanine and caffeine and hence, this later harvested material should be more marketable to functional food manufacturers, as they endeavor to address consumer desire for naturally-derived, functional ingredients. The best method to achieve improvements in quality and commercial value of Gyokuro-styled green tea grown on the Central Coast of New South Wales is to shade all visible light intensities to one per cent of ambient levels. The timing of this application should be carefully monitored as there appears to be an ideal relationship between the length of treatment and the minimum intensity of available light. Improvements can be observed in as little as two weeks when light intensity is reduced to one per cent. Shading treatments during the production of green tea are beneficial to produce high-quality products, in terms of maintaining their optimal quality for longer periods; however, shading treatments don't create better teas in terms of favorable qualities, compared to the 'First Harvest' teas, for example.

**Supplementary Materials:** The following supporting information can be downloaded at: https://www.mdpi.com/article/10.3390/beverages8020022/s1. Table S1: The total concentration of target constituents (mean ± SD) of green tea plants grown under various ambient light intensities; Table S2: The total concentration of target constituents (mean ± SD) of green tea plants grown with and without red-colored wavelengths compared to full ambient light conditions; Table S3: The total concentration of target constituents (mean ± SD) of green tea plants grown in reduced light intensity (90% opacity) compared to full ambient sunlight conditions over 11 weeks.

**Author Contributions:** Conceptualization, J.K. and M.A.K.; Data curation, J.K.; Formal analysis, J.K. and M.A.K.; Methodology, J.K. and M.A.K.; Validation, J.K. and M.A.K.; Writing—original draft, J.K.; Writing—review and editing, J.K. and M.A.K. All authors have read and agreed to the published version of the manuscript.

**Funding:** This research received no external funding.

**Institutional Review Board Statement:** Not applicable.

**Informed Consent Statement:** Not applicable.

**Data Availability Statement:** The data presented in this study are available on request from the corresponding author.

**Acknowledgments:** We greatly acknowledge the work of the New South Wales Department of Industry and Investment—Primary Industries Central Coast NSW and Kunitaro Coffee and Tea Company for completing the harvesting and processing of the green tea samples for this project.

**Conflicts of Interest:** J.K. currently works as the Innovation Manager at the Food Innovations Australia (FIAL), and the described project was completed at the University of Newcastle before his employment at FIAL. The authors declare no conflict of interest.

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
