# Peer review of "Optimizing the Quality and Commercial Value of Gyokuro-Styled Green Tea Grown in Australia"

_beverages, doi:10.3390/beverages8020022_

Round 1

Reviewer 1 Report

The study aimed to determine the optimal growing conditions of Gyokuro-styled tea grown on the New South Wales Central Coast of Australia, which included optimization of shade intensity, solar intensity, and duration of shade treatment. The purpose was to obtain tea of high quality. Moreover, the study brought some information on functional properties associated with optimizing the production of Gyokuro-styled green tea in Australia. Presented results may be of interest of other researchers.

The obtained results indicate that the application of shading is an effective method of creating teas containing a higher level of theanine. A higher level of theanine favours umami flavour, which may be desired by some producers and consumers.

The authors used the HPLC method for the determination of theanine, caffeine and catechins. However, even though the authors provided the reference for the HPLC method applied in the study, it would be useful to have a brief description with some information on the chromatographic conditions of the method.

In my opinion discussion on the results should be more elaborated.

Author Response

Response to Reviewer 1 Comments

Point 1: The authors used the HPLC method for the determination of theanine, caffeine and catechins. However, even though the authors provided the reference for the HPLC method applied in the study, it would be useful to have a brief description with some information on the chromatographic conditions of the method.

Response 1: Thank you for this feedback. We have endeavoured to incorporate more detail regarding the HPLC method and chromatographic conditions used in this study. However, we are cautious not to repeat too much of what has previously been published in our most recent paper (reference 19). Please see tracked changes in section 2.4.

Point 2: In my opinion discussion on the results should be more elaborated.

Response 2: The authors have reviewed and expanded components within the discussion section, also in response to feedback received from the other Reviewers.

Reviewer 2 Report

This manuscript explored the shading effect on green tea cultivation in Australia and used the previous results to evaluate the tea quality. The concept is clear and concise. There are some suggestions for authors and may assist to improve the MS more informative.

The instrument and the measuring method for the light intensity should be described and provide the manufacturer information. The photo proton flux density of each shading treatment is suggested to provide. 

More discussion about the result of the red-colored shade treatment is suggested.   

Shading increases the umami flavor but also reduces the content of total catechins. Catechins are also the key element for green tea as a functional food. There should be more evidence to support the increasing functional potential concept.

Author Response

Response to Reviewer 2 Comments

Point 1: The instrument and the measuring method for the light intensity should be described and provide the manufacturer information.

Response 1: A description of the instrument and measuring method for light intensity has now been included in the method section. We have also added some detail within the results section. The authors felt it also necessary to make note that a limitation of this study was that the light intensity was only measured at commencement. Please refer to section 2.2, Table 1 and 2 and the discussion (line 254).

Point 2: The photo proton flux density of each shading treatment is suggested to provide.

Response 2: This measure was not collected for this study.

Point 3: More discussion about the result of the red-coloured shade treatment is suggested.

Response 3: Thank you for this observation. In response, further detail about the red-coloured shading has been incorporated throughout the discussion section. Apologies for this oversight.

Point 4: Shading increases the umami flavour but also reduces the content of total catechins. Catechins are also the key element for green tea as a functional food. There should be more evidence to support the increasing functional potential concept.

Response 4: Additional context about the functional food properties/opportunities of green tea catechins have been included in the discussion section, which has been significantly revised.

Reviewer 3 Report

The authors investigated the impact of shading treatments on chemical composition and commercial value of Gyokuro-styled green tea in Australia. It is an interesting work, with the involvement of commercial value indices of tea products. However, some issues need to be addressed before considering publication in Beverages.

  1. Please give an explanation about the Scientific foundation of several quality markers in the present work, such as EGCG:EGC, EGCG:GCG. What is the meaning of these quality markers? Are they related with bitterness or astringency of tea?
  2. It is necessary to give the contents of all the chemicals detected in the study.
  3. Please give an explanation about the great increase of TC in red net-treated tea sample.
  4. Shading treatment is also a conventional agronomic operation of tea cultivation in China, and there are many publications about the effect of shading treatment on the chemical composition of sensory quality-related compounds. The recent studies were not introduced in the introduction section, which needs to be improved.

Author Response

Response to Reviewer 3 Comments

Point 1: Please explain the Scientific foundation of several quality markers in the present work, such as EGCG:EGC, EGCG:GCG. What is the meaning of these quality markers? Are they related with bitterness or astringency of tea?

Response 1: Thank you for this feedback. While we refer to our validation paper that details the relationship and value of these constituents as quality markers and value, we appreciate that this link was missing. In the introduction, we have added some detail about the importance of these ratios and markers in measuring the quality of green tea.

Point 2: It is necessary to give the contents of all the chemicals detected in the study.

Response 2: The contents of all chemicals detected in this study are noted in the methods section.

Point 3: Please give an explanation about the great increase of TC in red net-treated tea sample.

Response 3: We have included further detail regarding the increase in total catechin following red-coloured shading in the results and discussion section.

Point 4: Shading treatment is also a conventional agronomic operation of tea cultivation in China, and there are many publications about the effect of shading treatment on the chemical composition of sensory quality-related compounds. The recent studies were not introduced in the introduction section, which needs to be improved.

Response 4: Thank you for this suggestion. We have reviewed our introduction and have added additional details as suggested. Of note, we included one additional study, recently completed in the USA, and an emerging (Japanese-styled) green tea growing region, similar to that of our study on Australia.

Round 2

Reviewer 3 Report

The author still didn't fully address the issues raised before.  Authors need to provide the data about contents of the detected compounds. 

  1. Please give an explanation about the Scientific foundation of several quality markers in the present work, such as EGCG:EGC, EGCG:GCG. What is the meaning of these quality markers? Are they related with bitterness or astringency of tea?
  2. It is necessary to give the contents of all the chemicals detected in the study.

Author Response

Thank you for the further clarification of this feedback.

The authors have now included the total concentration data for all target constituents measured in this study. This includes theanine, caffeine and each of the catechins. These have been supplied for all study treatments and controls, as Supplementary Material.

Initially, we did not feel it was necessary to include this data as part of the full manuscript as it was used to calculate the quality markers and quality indicators which are the main indicators of this work. However, on reflection, we see this information as supplementary material is helpful to give context to the paper.

Please find reference to these supplementary tables, and where necessary, further explanation at lines: 210, 222, 223-226, 235.

In lieu of what would have been a rather large table of data for the green tea plants grown in reduced light intensity compared to full ambient sunlight conditions over 11 weeks, we have added in Figure 1 (page 6). This figure shows the change in quality markers over time.